# Integrative Metabolome and Proteome Analysis of Cerebrospinal Fluid in Parkinson’s Disease

**DOI:** 10.3390/ijms252111406

**Published:** 2024-10-23

**Authors:** Seok Gi Kim, Ji Su Hwang, Nimisha Pradeep George, Yong Eun Jang, Minjun Kwon, Sang Seop Lee, Gwang Lee

**Affiliations:** 1Department of Molecular Science and Technology, Ajou University, Suwon 16499, Republic of Korea; 2Department of Physiology, Ajou University School of Medicine, Suwon 16499, Republic of Korea; 3Department of Pharmacology, Inje University College of Medicine, Busan 50834, Republic of Korea

**Keywords:** cerebrospinal fluid, integrated omics, metabolomics, Parkinson’s disease, proteomics

## Abstract

Parkinson’s disease (PD) is a common neurodegenerative disorder characterized by the loss of dopaminergic neurons in the substantia nigra. Recent studies have highlighted the significant role of cerebrospinal fluid (CSF) in reflecting pathophysiological PD brain conditions by analyzing the components of CSF. Based on the published literature, we created a single network with altered metabolites in the CSF of patients with PD. We analyzed biological functions related to the transmembrane of mitochondria, respiration of mitochondria, neurodegeneration, and PD using a bioinformatics tool. As the proteome reflects phenotypes, we collected proteome data based on published papers, and the biological function of the single network showed similarities with that of the metabolomic network. Then, we analyzed the single network of integrated metabolome and proteome. In silico predictions based on the single network with integrated metabolomics and proteomics showed that neurodegeneration and PD were predicted to be activated. In contrast, mitochondrial transmembrane activity and respiration were predicted to be suppressed in the CSF of patients with PD. This review underscores the importance of integrated omics analyses in deciphering PD’s complex biochemical networks underlying neurodegeneration.

## 1. Introduction

Parkinson’s disease (PD) is the second most common neurodegenerative disorder characterized by the progressive loss of dopaminergic neurons, and understanding its underlying molecular mechanisms remains a key challenge in developing effective therapeutic strategies [1,2]. This neuronal loss significantly reduces dopamine levels in the striatum, leading to motor symptoms, including tremor, rigidity, bradykinesia, and postural instability, as well as non-motor symptoms, such as cognitive decline and autonomic dysfunction [3,4]. Among the theories of PD pathogenesis, mitochondrial dysfunction, including impaired mitochondrial respiration, DNA mutation, or dynamics, has emerged as a potential cause of PD [5,6,7,8]. It has been revealed that mitochondrial dysfunction is connected to PD, but the exact mechanism remains unclear.

The cerebrospinal fluid (CSF) provides distinct advantages for diagnosing PD due to its direct contact with the brain, allowing it to accurately reflect the pathophysiological conditions in the central nervous system [9,10,11,12]. As the CSF is in direct contact with the brain, it contains proteins, metabolites, and other molecular markers that are more representative of neurodegenerative processes than are peripheral fluids like blood. Among omics data, a meta-analysis of CSF metabolomes based on bioinformatics suggested a more quantitative understanding of the pathological mechanisms underlying PD symptoms and provided omics approaches for identifying biomarkers [13,14]. However, no critical CSF biomarkers for PD have been established yet.

Metabolic analysis has proven to be a powerful tool for elucidating the pathological mechanisms of PD [10,14,15,16,17,18,19,20], as metabolite levels are altered under pathological conditions, serving as biochemical and physiological indicators. Metabolomics approaches have facilitated the discovery of novel metabolic biomarkers for diagnosing the complex symptoms of PD, particularly in CSF, which is the most relevant biological fluid due to its proximity to the brain and its reflection of the disease’s pathophysiological state [21]. For example, the metabolic profiling of free fatty acids and polyamine in CSF was helpful in distinguishing between PD and multiple system atrophy (MSA), aiding in understanding the disease mechanisms and uncovering clinically relevant biomarkers [22,23]. Such studies are helpful in classifying PD and MSA for suitable treatment at the early stage of the disease. In particular, focusing on concentration ratios and metabolite profiling, known as metabotyping [24], can help distinguish between PD and MSA. Therefore, metabolomics provides valuable insights into these biochemical alterations, enabling researchers to gain a deeper understanding of PD’s progression and identify potential biomarkers for early diagnosis.

Although the CSF metabolome reflects the phenotype of PD well, there are some limitations, as metabolome data alone do not accurately represent the exact biochemical pathways. Hence, additional omics analyses, such as proteome and microRNAs, can be performed to compensate for metabolomic analysis. Among omics, proteomics, the large-scale study of proteins, has been studied as a valuable approach for analyzing CSF in PD [19,25,26,27,28,29]. Although approximately 20% of proteins in CSF are derived from brain cells, such as neurons, astrocytes, and glial cells, with approximately 80% derived from peripheral blood filtration [11,30], CSF proteins still provide a unique window into the protein alterations associated with neurodegeneration. By identifying and quantifying dysregulated proteins in PD, proteomics can elucidate neurodegenerative mechanisms and uncover potential biomarkers that reflect disease progression, neuronal damage, and other pathological processes. Proteomic analysis of CSF has already revealed several candidate proteins involved in synaptic dysfunction, inflammation, autophagy–lysosomal dysfunction, mitochondrial dysfunction, and oxidative stress, which are key to understanding PD [31,32]. This approach enhances our understanding of PD pathophysiology and aids in the development of diagnostic and therapeutic strategies by identifying protein signatures specific to PD.

The combination of metabolomics and proteomics to analyze the CSF of patients with PD provides a comprehensive approach to understanding the complex biochemical and molecular changes [19]. By combining these two omics approaches, researchers can simultaneously analyze small metabolites and proteins, offering a more comprehensive view of the pathological processes involved [33,34,35]. This integration allows for the identification of correlations between metabolic and protein alterations, which can lead to more accurate biomarker discovery. Additionally, it enhances the ability to detect early changes in disease states, improves the understanding of disease mechanisms, and supports the development of targeted therapeutic interventions by providing a multi-layered perspective on cellular dysfunction. In this review, we collected metabolomic and proteomic data on the CSF of patients with PD from published literature until now and integrated them with bioinformatics and in silico prediction. This approach proved especially powerful for unraveling the intricate network of biological events in the CSF of patients with PD. Additionally, CSF-based biomarkers can provide valuable insights into disease progression and enable the early detection of PD, which is crucial for timely intervention and treatment strategies. Therefore, integrated omics contribute to more precise evaluations of PD symptoms. In this review, we discuss the following: (i) the diagnosis of PD through CSF, (ii) metabolomics approaches in CSF of patients with PD, (iii) proteomics approaches in CSF of patients with PD, and (iv) the integration of metabolomics and proteomics in CSF of patients with PD for elucidating the mechanisms and discovering specific biomarkers for PD.

## 2. Diagnosis of PD Through CSF

CSF is essential for normal brain function, including protection, supplementation of nutrients, and waste removal [36]. Its changes can reflect a variety of neurological symptoms and signs [37]. CSF is secreted by ependymal cells in the choroid plexus located within the brain’s ventricles [38]. It serves as a key biofluid for biomarker discovery [39], reflecting pathophysiological conditions in the brain.

Altered CSF metabolite profiles, such as free fatty acids and polyamine, hold the potential for distinguishing PD from MSA [22,23], which could improve early diagnostic accuracy for PD. In addition, bioinformatics analysis of metabolites in the CSF from patients with PD has provided important insights into the disease’s pathophysiology, revealing disruptions in metabolic pathways related to nervous system disorders, inflammation, ATP reduction, and neurodegenerative diseases [13]. Notably, specific metabolites, such as amino acids, free fatty acids, polyamines, and dopamine, are altered in the CSF from patients with PD [10,13,40,41,42], reflecting both neurodegeneration and compensatory mechanisms. Standardizing metabolomic techniques and validating candidate biomarkers in larger, more diverse populations are essential for advancing the diagnostic and therapeutic landscape in PD. Despite much research, the precise role of these metabolites in disease progression and their diagnostic utility are still being explored.

Nowadays, there are limitations in using CSF-based biomarkers to differentiate between PD and other neurodegenerative diseases, such as MSA. Most analytical approaches for detecting metabolites in CSF are target-based, using NMR and GC-MS. These methods also face challenges, as molecules below nanomolar concentrations can hardly be detected. Moreover, there are significant constraints in the multi-dimensional bioinformatic analysis of proteins and metabolites, and the related software has not yet been fully developed. To overcome these limitations, non-targeted approaches are being employed, which allow for the detection of a broader range of novel molecules. Advanced technologies, such as nanopore sensing and surface-enhanced Raman spectroscopy [43,44], which can detect molecules at picomolar and femtomolar concentrations, are also being integrated into these analyses. Finally, as large datasets are obtained, a new challenge arises in extracting multidimensional molecular data through dimensional reduction and analyzing them using various artificial intelligence (AI) algorithms. These analyses will be based on networks that accurately reflect neurodegenerative disease-specific molecular pathways, facilitating the identification of biomarkers to differentiate between PD and other neurodegenerative diseases. This is briefly discussed in Section 5: Integration of Metabolomics and Proteomics in CSF of Patients with PD.

## 3. Metabolomics Approaches in CSF of Patients with PD

Metabolites, as intermediates or products of cell metabolism, are typically classified as low-molecular-weight compounds (less than 1.5 kDa in size) [45], with approximately 2500 cataloged in humans [46]. Metabolomics, the quantitative analysis of the multiparametric metabolic response to pathological stimuli in living organs, provides a comprehensive view of metabolic changes. Clinically, metabolomics offers a unique position over other omics technologies by directly reflecting real-time biological processes such as cellular signaling, energy metabolism, and enzyme-mediated biochemical reactions [47]. This unique ability to capture real-time biological processes makes it a valuable tool in understanding disease mechanisms, monitoring treatment, and predicting diagnosis, particularly in neurodegenerative conditions such as Alzheimer’s disease, Huntington’s disease, and PD [48,49].

In analytic practice, the most commonly employed analytical tools for metabolic profiling include nuclear magnetic resonance (NMR) spectroscopy, gas chromatography-mass spectrometry (GC-MS), and liquid chromatography with tandem mass spectrometry (LC-MS/MS). NMR, while having a relatively lower sensitivity (>1 nmol) and resolution compared with those of GC-MS or LC-MS/MS [50], offers several clinical advantages, including high reproducibility, minimal sample preparation, and lower per-sample costs [51,52]. Recent innovations, such as cryogenic NMR spectroscopy, have further improved its sensitivity in NMR experiments [53] and have been applied to analyze the CSF in patients with PD [54,55,56], making it more applicable to clinical metabolomic studies. Mass spectrometry (MS)-based metabolomics, with its high sensitivity and accuracy, is frequently used in clinical settings in combination with separation techniques, such as liquid chromatography (LC), gas chromatography (GC), and capillary electrophoresis (CE), allowing for more precise detection of metabolites [57,58]. GC-MS is particularly valuable for identifying and profiling PD-specific metabolic signatures in CSF [22,23,59,60,61], which can be used for early diagnosis and prognostic assessment. The development of analytical instruments will increase accuracy and sensitivity, enhancing the understanding of underlying mechanisms and the diagnosis of PD through CSF.

We obtained the list of metabolites that were altered in the CSF of patients with PD using various analytical methods, including NMR, GC-MS, LC-MS/MS, and meta-analyses, to understand PD pathologies through CSF (Table 1). The changes in the levels of amino acids, nucleotides, organic acids, fatty acids, polyamines, phenolic compounds, and their derivatives were identified in the CSF of patients with PD. The metabolomic network was then constructed using ingenuity pathway analysis (IPA, http://www.ingenuity.com, accessed on 19 September 2024) [62] to identify the relationships between metabolites and biological functions and diseases (Figure 1A). IPA is a powerful bioinformatics tool that enables the interpretation of complex omics data by identifying relevant biological pathways and networks, providing deep insights into disease mechanisms and therapeutic targets. Its extensive curated database makes it invaluable for transforming large datasets into interpretable biological knowledge. Among the 54 metabolites, 23 increased and 27 decreased metabolites were uploaded, excluding four components (L-prolyl-L-tyrosine, 7α,(25R)26-dihydroxycholesterol, 7α,x,y-trihydroxycholest-4-en-3-one, and 2(N)-methyl-norsalsolinol) because these components were not included in the IPA program. Through the metabolome network, we confirmed the connections between metabolites and those between metabolites and four biological functions and diseases (mitochondrial transmembrane potential, mitochondrial respiration, neurodegeneration, and PD). Based on the increased/decreased metabolites, the activation or inhibition of functions and diseases was predicted (Figure 1B). The network predicted mitochondrial dysfunction and the development of neurodegeneration and PD. Decreased levels of betaine, inosine, L-glutamic acid, levodopa, spermidine, and taurine and increased levels of 4-hydroxynonenal, arachidonic acid, N-methyl-(R)-salsolinol, and nitric oxide appeared to directly impact mitochondrial functions or contribute to diseases. Particularly, decreased inosine levels and increased nitric oxide levels were emphasized as having adverse effects on mitochondrial function and diseases. Inosine, a type of purine, has been reported to increase mitochondrial respiration in cells under stressful conditions [63,64], and inosine treatment has been demonstrated to have a neuroprotective function in models of PD [65,66]. Additionally, a phase 3 clinical trial was conducted to treat PD through inosine intake (NCT02642393). Nitric oxide, despite its essential role in several signaling pathways, can cause oxidative stress and directly inhibit the mitochondrial respiratory chain [67,68]. Several mechanisms by which nitric oxide contributes to neurotoxicity, particularly in PD, have been proposed, including axo-dendritic dysfunction, mitochondrial dysfunction, and disruptions in dopamine homeostasis [69,70]. The metabolomic network suggests that mitochondrial function and PD are interrelated through the metabolome.

## 4. Proteomics Approaches in CSF of Patients with PD

Proteomics has been used to investigate neurodegenerative diseases [91,92] because it facilitates the understanding of molecular processes, protein compositions, posttranslational modifications, and changes across heterogeneous conditions [93]. Traditional techniques for analyzing proteins in samples include two-dimensional gel electrophoresis [26], immunology multiplex assay [94], and enzyme-linked immunosorbent assay (ELISA) [95]. In addition, LC-MS/MS [96,97,98] has been employed as an advanced analytical method to overcome the limitations of classical techniques and provide more precise quantification and identification of proteins in CSF of patients with PD. These techniques allow for a comprehensive investigation into protein alterations associated with PD, thus helping us understand molecular processes in PD and identify potential biomarkers.

Proteomic profiles were obtained from the CSF of patients with PD to provide additional information following the metabolomic analysis, revealing 38 increased and 38 decreased proteins (Table 2). Similar to the metabolomic analysis, the proteomic network was constructed using IPA, and relationships among proteins, biological functions, and diseases were confirmed (Figure 2A). The prediction of four functions and diseases showed that inhibition of mitochondrial transmembrane potential and respiration, as well as activation of neurodegeneration and PD, were expected, consistent with the findings from the metabolomic network (Figure 2B). Among the identified proteins, 11 proteins (APOE, CO3, IL1B, IL4, IL6, NGF, PARK7, TGFA, TGFB1, TNFA, and VEGFA) directly affected mitochondrial functions or diseases, highlighting the increased levels of IL1B and decreased levels of NGF, which have adverse effects on both mitochondrial functions and related diseases. IL1B (interleukin 1 beta) is a proinflammatory cytokine involved in inflammatory responses from a wide range of stimuli, and its treatment has been reported to inhibit mitochondrial activity and function [99,100,101]. Koprich et al. confirmed that neuroinflammation can exacerbate the disease in a PD model, and that treatment with an IL1B receptor antagonist can reduce the loss of dopaminergic neurons [102]. Nerve growth factor (NGF) is a neurotrophin that plays a key role in various processes such as neuronal survival, differentiation, and maturation [103]. A decrease in NGF level has been observed in the serum of PD models [104,105], and NGF infusion has been pursued in a single PD patient to support adrenal chromaffin tissue engrafted in the patient’s putamen [106,107]. The effects of NGF on mitochondrial function and dynamics support the multifaceted nature of NGF in relation to mitochondria and PD [108,109]. Analysis of the proteomic network suggests that mitochondrial function is linked to PD and that proteomic changes in CSF play an important role in understanding its pathology.

## 5. Integration of Metabolomic and Proteomics in CSF of Patients with PD

The metabolic profile serves as a direct endpoint of biological metabolism, closely reflecting the associated phenotype, making it physiologically significant for understanding neurodegenerative disease states [118,119,120]. However, due to the inability to amplify metabolites, metabolomics inherently has lower coverage compared to that of transcriptomics and proteomics [121]. Moreover, as metabolites are the end products of biological activities, information on cellular and molecular mechanisms is deficient. Integrating multiple omics technologies is highly recommended in neurodegenerative disease research to mitigate these limitations, although developing such strategies remains challenging due to multi-staged analysis and meta-dimensional analysis [122,123]. Recently, integrated omics approaches in PD research have uncovered novel biological changes underlying various conditions in vitro, providing potential clinical insights [124,125,126,127,128]. This comprehensive analysis with integrated omics can lead to the discovery of novel biomarkers and therapeutic targets, ultimately enhancing clinical applications.

To integrate the omics data and analyze the metabolome and proteome simultaneously, each metabolome and proteome profile was integrated into IPA, and correlations between altered metabolites and proteins in the CSF from patients with PD were identified (Figure 3A). The integrated network underscores the connectivity among nitric oxide, IL1B, and NGF among the molecules highlighted in the single omics analyses. The proportional relationship between IL1B and nitric oxide has been proven through studies demonstrating that nitric oxide production is induced by IL1B treatment or that both IL1B and nitric oxide levels increase concurrently in response to an inflammatory response [129,130]. In contrast, NGF and nitric oxide have an inverse proportion, as supported by the research indicating that nitric oxide suppresses NGF production [131]. Subsequently, integrated omics predicted four biological functions and diseases, showing similar prediction tendencies to each single omics prediction (inhibition of mitochondrial functions and activation of neurodegeneration and PD) (Figure 3B). In addition, using integrated omics, analyses of canonical pathways, diseases, and biological functions were performed through IPA (Figure 3C,D). The top seven activated/inhibited canonical pathways showed that activation of the immune response was predicted, along with potential issues related to molecular transportation (Figure 3C). In particular, various inflammatory and immune-related pathways were predicted to be activated, with molecules such as IL1B, IL4, IL6, TNF, and nitric oxide appearing to play significant roles (Appendix A). These observations reflect the activation of multiple immune response pathways in PD. Among the top seven activated/inhibited diseases and biological functions, vascular endothelial cell damage and immune cell responses were predicted to be activated. Conversely, mitochondrial and neuronal cell functions were expected to be inhibited in patients with PD (Figure 3D, Appendix A). The integration of the metabolome and proteome of CSF from patients with PD suggests that this approach provides broader insights than do individual omics data in understanding PD. Furthermore, the integration of additional omics data may expand our understanding of the complicated pathophysiology of PD.

MicroRNAs in the CSF of patients with PD are promising targets for analysis and integration into multi-omics. MicroRNAs are a class of small, non-coding RNAs, approximately 20 nucleotides in length, that interfere with the gene expression of target mRNA by binding to the 3′ untranslated region [132]. These small RNAs have been extensively studied as precise recognition and clinical biomarkers in various diseases, including neurodegenerative conditions such as PD [133,134,135]. In biological fluids, including cell-free CSF in PD, microRNAs have emerged as promising and potential biomarkers [136,137,138,139,140]. Several studies have suggested and are continuing to investigate microRNAs that are significantly differentially expressed in the CSF of patients with PD [13,139,141,142]. For example, Tan et al. found 21 upregulated microRNAs in the early stage of PD and suggested miR-409-3p as an indicator of early PD [141]. Burgos et al. found 17 microRNAs differentially expressed in the CSF of PD patients, highlighting five microRNAs (miR-127-3p, miR-443, miR-431-3p, miR-136-3p, and miR-10a-5p) commonly differentially expressed in both PD and Alzheimer’s disease compared to control subjects [142]. Additionally, we obtained a list of microRNAs associated with PD from the IPA database [143,144] (Appendix A). Although efforts to find microRNA markers in CSF are ongoing, the underlying mechanisms of microRNAs in PD remain poorly understood. If further studies explore microRNAs in greater detail, they could serve as promising candidates for integration with other omics approaches, providing deeper insights into the pathophysiology of PD and enhancing biomarker discovery.

As integrated omics data constitute big data, computational approaches and artificial intelligence (AI) must be considered for handling these datasets [145]. Among AI, machine learning (ML) is a powerful tool for managing large-scale datasets, including those derived from integrated omics data [146,147]. ML can be broadly classified into supervised, unsupervised, and reinforcement learning [148]. Supervised learning algorithms, such as support vector machines, K-nearest neighbors, random forest, and decision trees, are commonly used [149]. In contrast, unsupervised learning methods work with unlabeled data, grouping them based on similarity. Unsupervised learning algorithms include K-means, ordering points to identify the clustering structure, density-based spatial clustering of applications with noise, and hierarchical density-based spatial clustering of applications with noise [150]. Additionally, principal component analysis, t-distributed stochastic neighbor embedding, and uniform manifold approximation and projection assist in data analysis by reducing the dimensionality of large datasets while preserving important relationships [151]. Applying these clustering algorithms allows multi-omics datasets to be dimensionally reduced and organized for unsupervised learning. In fact, these approaches are being used for multi-omics analyses in various fields [152,153,154,155]. The use of ML algorithms in integrated omics data analysis—incorporating metabolomic and proteomic datasets from the CSF of patients with PD—has the potential to reduce biases, enhance clinical data interpretation, and support the development of precision medicine approaches for this neurodegenerative disease. Furthermore, by analyzing large datasets, ML can uncover subtle patterns and aid in identifying biomarker candidates for PD by detecting of omics-based alteration patterns, thereby advancing the development of reliable biomarkers. In summary, the integrated omics approach provides a holistic view of biological processes, facilitating the discovery of novel biomarkers, improving disease classification, and advancing precision medicine strategies. Its combination with ML further enhances its potential by allowing for managing and interpreting complex multi-dimensional data, as demonstrated in recent PD studies.

This review has several limitations: (i) The samples and analytical methods were not standardized, as we derived significantly changed metabolites and proteins from independent studies based on the existing literature. (ii) There is limited information regarding the mechanisms of microRNAs in PD, which precludes a comprehensive examination of single-omics and integrated-omics analyses. The combination of well-controlled patient samples, standardized analytical methods, and an accumulation of mechanistic studies involving microRNAs and other omics could substantially enhance our understanding of CSF changes in PD.

## 6. Conclusions and Future Directions

This review highlights the critical role of CSF in reflecting metabolic and proteomic alterations associated with PD. The combined metabolomic and proteomic data analysis provides a more comprehensive understanding of PD pathophysiology, particularly concerning mitochondrial dysfunction and neurodegeneration. Our bioinformatics analysis of a single network combining both metabolome and proteome data revealed distinct biological functions, with neurodegeneration and PD-related processes predicted to be activated, while mitochondrial transmembrane transport and respiration were suppressed. These findings reinforce the importance of multi-omics approaches in elucidating the intricate biochemical networks involved in CSF of patients with PD. Future research should focus on expanding the scope of detectable metabolites and integrating additional omics data to further unravel the mechanisms underlying PD and explore new therapeutic targets.

## Figures and Tables

**Figure 1 ijms-25-11406-f001:**
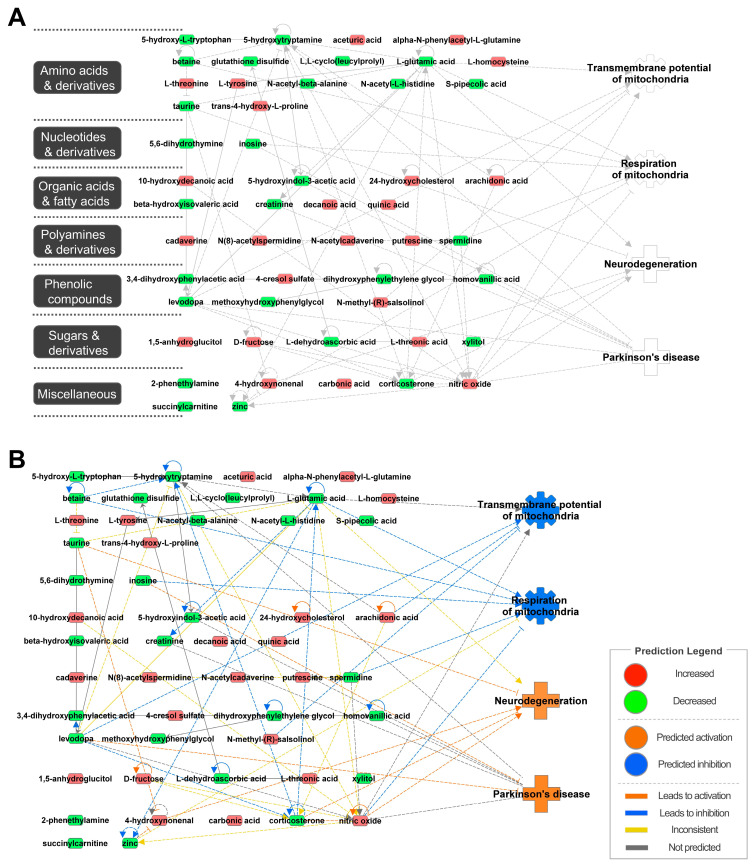
Biological functions and disease-related metabolomic network from ingenuity pathway analysis (IPA). (**A**) Network of metabolites and associated functions and diseases in the cerebrospinal fluid of patients with Parkinson’s disease. (**B**) Prediction of the metabolomic network through the altered level of metabolites. Red and green indicate increased and decreased metabolites in the CSF of PD patients, respectively. Orange and blue represent the activation and inhibition of metabolites, functions, and diseases. Solid and dotted line represents direct and indirect relationships, respectively.

**Figure 2 ijms-25-11406-f002:**
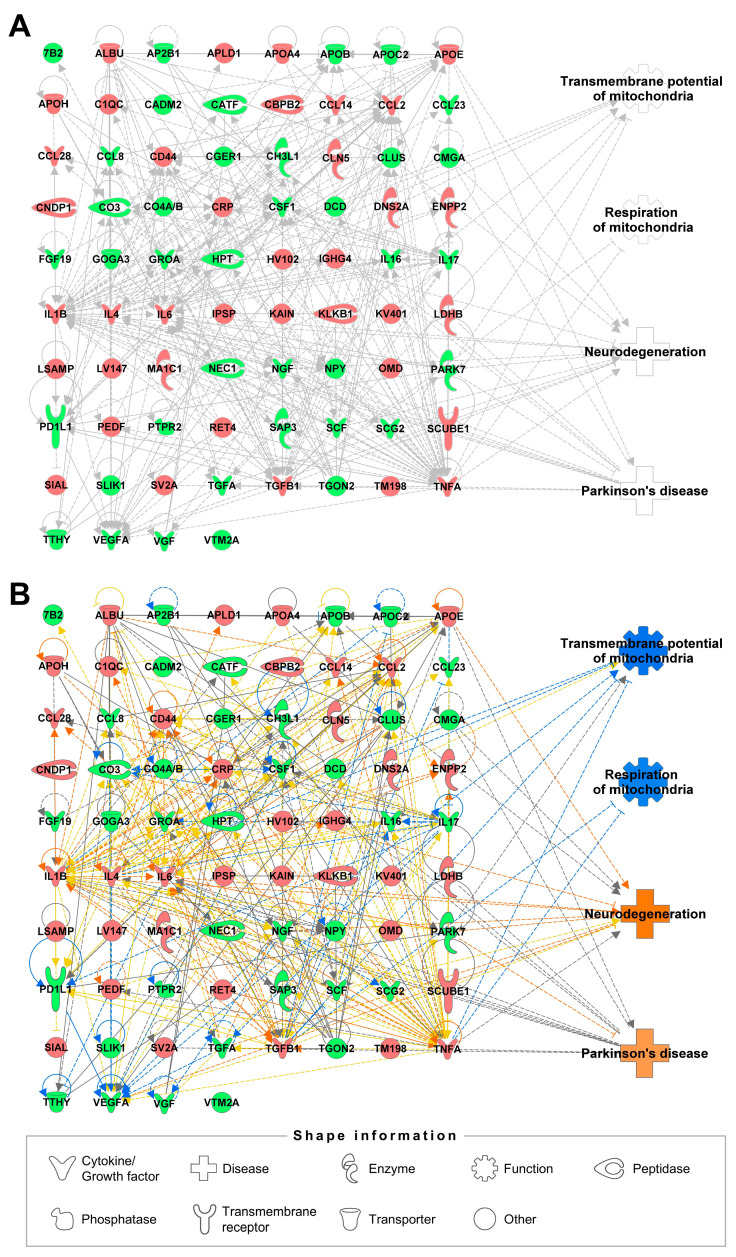
Biological functions and disease-related proteomic network from ingenuity pathway analysis (IPA). (**A**) Network of proteins and associated functions and diseases in the cerebrospinal fluid of patients with Parkinson’s disease. (**B**) Prediction of the proteomic network through the altered level of proteins. Increased levels of proteins are represented in red, while decreased levels are represented in green. Blue indicates the predicted inhibition, and orange shows the expected activation of cellular functions or diseases. Solid and dotted line represents direct and indirect relationships, respectively.

**Figure 3 ijms-25-11406-f003:**
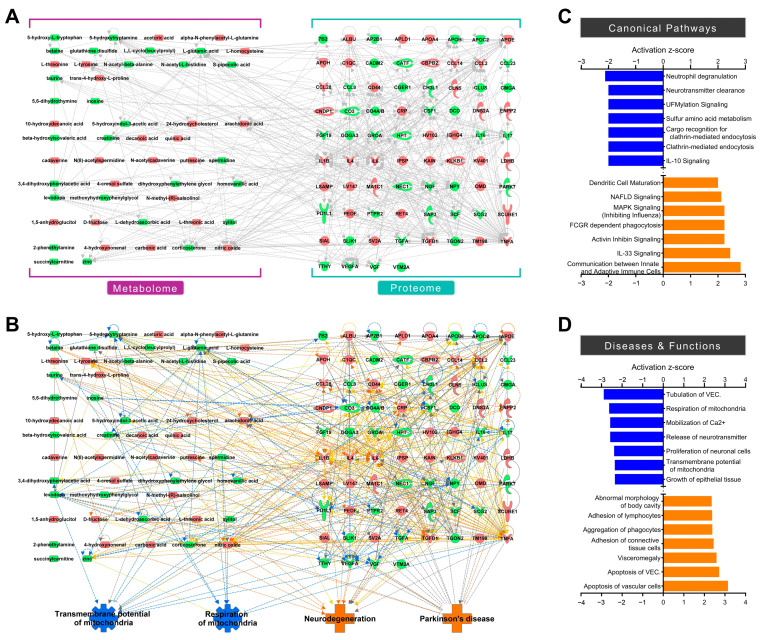
Integrative omics network prediction and omics analyses from ingenuity pathway analysis (IPA). (**A**) Integrative omics network with metabolome and proteome. (**B**) Prediction of the integrative omics network with cellular functions and diseases. Red and green indicate increased and decreased molecules in the CSF of PD patients, respectively. Orange and blue represent the activation and inhibition of molecules, functions, and diseases. Solid and dotted line represents direct and indirect relationships, respectively. (**C**,**D**) Integrative omics analyses of the top seven activated/inhibited canonical pathways (**C**) and diseases and biological functions (**D**). A negative or positive activation z-score indicates inhibition or activation, respectively. VEC: vascular endothelial cell.

**Table 1 ijms-25-11406-t001:** List of altered metabolites in the cerebrospinal fluid of patients with Parkinson’s disease.

Metabolite	Increased/ Decreased	Analysis Method ^a^	References
<Amino Acids and Derivatives>			
5-hydroxy-L-tryptophan	Decreased	LC-ECD	[71]
5-hydroxytryptamine	Decreased	LC-ECD	
aceturic acid	Increased	LC-MS/MS and GC/MS	[15]
alpha-N-phenylacetyl-L-glutamine	Increased	LC-MS/MS	[41]
betaine	Decreased	LC-MS/MS and GC/MS	[15]
glutathione disulfide	Decreased	LC-MS/MS and GC/MS	
L,L-cyclo(leucylprolyl)	Decreased	LC-MS/MS and GC/MS	
L-glutamic acid	Decreased	Meta-analysis	[72]
L-homocysteine	Increased	LC-ECD	[73]
L-prolyl-L-tyrosine	Increased	LC-MS/MS	[41]
L-threonine	Increased	Meta-analysis	[72]
L-tyrosine	Increased	LC-MS/MS	[74]
		Meta-analysis	[72]
N-acetyl-beta-alanine	Decreased	LC-MS/MS and GC/MS	[15]
N-acetyl-L-histidine	Decreased	LC-MS/MS and GC/MS	
S-pipecolic acid	Decreased	LC-MS/MS and GC/MS	
taurine	Decreased	LC-ECD	[75]
		Meta-analysis	[72]
trans-4-hydroxy-L-proline	Increased	LC-MS/MS	[74]
<Nucleotides and Derivatives>			
5,6-dihydrothymine	Decreased	LC-MS/MS and GC/MS	[15]
inosine	Decreased	LC-MS/MS and GC/MS	
<Organic Acids and Fatty Acids>			
10-hydroxydecanoic acid	Increased	FT-ICR-MS	[76]
24-hydroxycholesterol	Increased	IDMS	[77]
5-hydroxyindol-3-acetic acid	Decreased	ELISA	[17]
7α,(25R)26-dihydroxycholesterol	Increased	LC-MS	[78]
7α,x,y-trihydroxycholest-4-en-3-one	Increased	LC-MS	
arachidonic acid	Increased	FT-ICR-MS	[76]
beta-hydroxyisovaleric acid	Decreased	GC-MS	[79]
creatinine	Decreased	NMR	[54]
decanoic acid	Increased	FT-ICR-MS	[76]
quinic acid	Increased	FT-ICR-MS	
<Polyamines and Derivatives>			
cadaverine	Increased	GC-MS	[22]
N(8)-acetylspermidine	Increased	GC-MS	
N-acetylcadaverine	Increased	GC-MS	
putrescine	Increased	LC-MS/MS	[74]
		GC-MS	[22]
spermidine	Decreased	GC-MS	[22]
<Phenolic Compounds>			
2(N)-Methyl-norsalsolinol	Decreased	LC-ECD	[80]
3,4-dihydroxyphenylacetic acid	Decreased	LC-MS/MS LC-ECD	[18] [10,75,81]
4-cresol sulfate	Increased	LC-MS/MS FT-ICR-MS	[82] [76]
dihydroxyphenylethylene glycol	Decreased	LC-ECD	[10,81]
homovanillic acid	Decreased	LC-MS/MS LC-ECD	[18] [81,83]
levodopa	Decreased	LC-ECD	[10]
methoxyhydroxyphenylglycol	Decreased	LC-ECD	[81]
N-methyl-(R)-salsolinol	Increased	LC-ECD	[84,85]
<Sugars and Derivatives>			
1,5-anhydroglucitol	Increased	LC-MS/MS and GC/MS	[15]
D-fructose	Increased	GC-MS	[86]
L-dehydroascorbic acid	Decreased	GC-MS	
L-threonic acid	Increased	GC-MS	
xylitol	Decreased	GC-MS	[79]
<Miscellaneous>			
2-phenethylamine	Decreased	GC-MS	[87]
4-hydroxynonenal	Increased	GC-MS	[88]
carbonic acid	Increased	GC-MS	[86]
corticosterone	Decreased	LC-MS/MS and GC/MS	[15]
nitric oxide	Increased	Meta-analysis	[89]
succinylcarnitine	Decreased	LC-MS/MS and GC/MS	[15]
zinc	Decreased	Meta-analysis	[90]

^a^ Abbreviations: LC-ECD, liquid chromatography with electrochemical detection; LC-MS/MS, liquid chromatography with tandem mass spectrometry; GC-MS, gas chromatography with mass spectrometry; FT-ICR-MS, Fourier transform–ion cyclotron resonance–mass spectrometer; IDMS, isotope dilution mass spectrometry; ELISA, enzyme-linked immunosorbent assay; LC-MS, liquid chromatography with mass spectrometry; NMR, nuclear magnetic resonance.

**Table 2 ijms-25-11406-t002:** List of altered proteins in the cerebrospinal fluid of patients with Parkinson’s disease.

Protein Symbol ^a^	Accession Number ^b^	Increased/ Decreased	Analysis Method ^c^	References
7B2	P05408	Decreased	LC-MS/MS	[110]
ALBU	P02768	Increased	LC-MS/MS	[26]
AP2B1	P63010	Decreased	SPE and PRM-MS	[111]
APLD1	Q96LR9	Increased	LC-MS/MS	[110]
APOA4	P06727	Increased	LC-MS/MS	
APOB	P04114	Decreased	LC-MS/MS	[112]
APOC2	P02655	Decreased	LC-MS/MS	[110]
APOE	P02649	Increased	LC-MS/MS	[26]
APOH	P02749	Increased	LC-MS/MS	[113]
C1QC	P02747	Increased	LC-MS/MS	[114]
CADM2	Q8N3J6	Decreased	LC-MS/MS	[110]
CATF	Q9UBX1	Decreased	SPE and PRM-MS	[111]
CBPB2	Q96IY4	Increased	LC-MS/MS	[110]
CCL14	Q16627	Increased	Protein microarray and LC-MS/MS	[115]
CCL2	P13500	Increased	Meta-analysis	[89]
			Protein microarray	[116]
CCL23	P55773	Decreased	Systematic review	[89]
CCL28	Q9NRJ3	Increased	Systematic review	
CCL8	P80075	Decreased	Systematic review	
CD44	P16070	Increased	LC-MS/MS	[117]
CGER1	Q99674	Decreased	LC-MS/MS	[114]
CH3L1	P36222	Decreased	Meta-analysis	[90]
			Meta-analysis (ELISA)	[89]
CLN5	O75503	Increased	Protein microarray and LC-MS/MS	[115]
CLUS	P10909	Decreased	LC-MS/MS	[26]
CMGA	P10645	Decreased	LC-MS/MS	[110]
CNDP1	Q96KN2	Increased	LC-MS/MS	[26]
CO3	P01024	Decreased	LC-MS/MS	
CO4A/B	P0C0L4/P0C0L5	Decreased	LC-MS/MS	
CRP	P02741	Increased	Meta-analysis	[89]
CSF1	P09603	Decreased	Systematic review	
DCD	P81605	Decreased	LC-MS/MS	[26]
DNS2A	O00115	Increased	LC-MS/MS	[110]
ENPP2	Q13822	Increased	LC-MS/MS	[26]
FGF19	O95750	Decreased	Systematic review	[89]
GOGA3	Q08378	Decreased	LC-MS/MS	[112]
GROA	P09341	Decreased	Systematic review	[89]
HPT	P00738	Decreased	LC-MS/MS	[26]
HV102	P23083	Increased	LC-MS/MS	[110]
IGHG4	P01861	Increased	LC-MS/MS	
IL16	Q14005	Decreased	Systematic review	[89]
IL17	Q16552	Decreased	Systematic review	
IL1B	P01584	Increased	Meta-analysis	
IL4	P05112	Increased	Meta-analysis (ELISA)	
			Meta-analysis (multiplex cytokine)	
IL6	P05231	Increased	Meta-analysis	
IPSP	P05154	Increased	LC-MS/MS	[110]
KAIN	P29622	Increased	LC-MS/MS	
KLKB1	P03952	Increased	LC-MS/MS	
KV401	P06312	Increased	LC-MS/MS	
LDHB	P07195	Increased	LC-MS/MS	[26]
LSAMP	Q13449	Increased	LC-MS/MS	[113]
LV147	P01700	Increased	LC-MS/MS	[110]
MA1C1	Q9NR34	Increased	Protein microarray and LC-MS/MS	[115]
NEC1	P29120	Decreased	LC-MS/MS	[110]
NGF	P01138	Decreased	Systematic review	[89]
NPY	P01303	Decreased	LC-MS/MS	[110]
OMD	Q99983	Increased	Protein microarray and LC-MS/MS	[115]
			LC-MS/MS	[117]
PARK7	Q99497	Decreased	Meta-analysis	[90]
PD1L1	Q9NZQ7	Decreased	Systematic review	[89]
PEDF	P36955	Increased	LC-MS/MS	[26]
PTPR2	Q92932	Decreased	LC-MS/MS	[110,114]
RET4	P02753	Increased	LC-MS/MS	[110]
SAP3	P17900	Decreased	SPE and PRM-MS	[111]
SCF	P21583	Decreased	Systematic review	[89]
SCG2	P13521	Decreased	LC-MS/MS	[110]
SCUBE1	Q8IWY4	Increased	Protein microarray and LC-MS/MS	[115]
SIAL	P21815	Increased	Protein microarray	[116]
SLIK1	Q96PX8	Decreased	LC-MS/MS	[110]
SV2A	Q7L0J3	Increased	Protein microarray	[116]
TGFA	P01135	Decreased	Meta-analysis	[89]
TGFB1	P01137	Increased	Meta-analysis (ELISA)	
TGON2	O43493	Decreased	LC-MS/MS	[114]
TM198	Q66K66	Increased	LC-MS/MS	[110]
TNFA	P01375	Increased	Meta-analysis	[89]
TTHY	P02766	Decreased	LC-MS/MS	[26]
VEGFA	P15692	Decreased	Systematic review	[89]
VGF	O15240	Decreased	LC-MS/MS	[110,114]
VTM2A	Q8TAG5	Decreased	LC-MS/MS	

^a^ The protein symbol indicates the human protein assigned by UniProtKB. ^b^ The accession number indicates the unique identifier of the protein in UniProtKB. ^c^ Abbreviations: LC-MS/MS, liquid chromatography with tandem mass spectrometry; SPE, solid-phase extraction; PRM-MS, parallel reaction monitoring mass spectrometry; ELISA, enzyme-linked immunosorbent assay.

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
