# Peer review of "Integrative Metabolome and Proteome Analysis of Cerebrospinal Fluid in Parkinson’s Disease"

_ijms, 2024, doi:10.3390/ijms252111406_

Round 1

Reviewer 1 Report

Comments and Suggestions for Authors

The review presented by Kim et al. is interesting.

Can you provide more details on the size and diversity of the samples used for metabolomics and proteomics analysis? How do you account for the variability in cerebrospinal fluid (CSF) composition in different patients with Parkinson's disease?

What steps have been taken to validate the biomarkers identified by metabolomics and proteomics in larger, independent cohorts?

Considering that nuclear magnetic resonance (NMR) spectroscopy has a lower sensitivity compared to other methods such as mass spectrometry (MS), how did you ensure that all relevant metabolites were accurately detected?

How do you address the potential limitations of using in silico predictions in the study, particularly in predicting neurodegeneration and mitochondrial dysfunction?

While the integration of metabolomics and proteomics data is valuable, how did you deal with potential discrepancies between these datasets when you constructed the integrated network for Parkinson's disease?

Could you explain the specific bioinformatics tools that were used to correlate the metabolomic and proteomic data and how these ensure reliable prediction of cellular functions and disease states?

The study predicts inhibition of mitochondrial functions and activation of neurodegeneration based on altered metabolite and protein levels. Can you provide experimental evidence that directly links these predictions to the observed pathological processes in PD?

What impact do the observed changes in specific metabolites (such as inosine, levodopa and nitric oxide) have on the clinical course of Parkinson's disease?

What are the next steps to validate the identified biomarkers for clinical use and how do you plan to address potential challenges in moving from research findings to practical diagnostic tools?

Can you explain any known limitations or challenges in using CSF-based biomarkers to differentiate between Parkinson's disease and other neurodegenerative diseases, such as multiple system atrophy (MSA)?

Given the advances in analytical techniques, why were certain methods such as gas chromatography-mass spectrometry (GC-MS) chosen over others for metabolome profiling? Were other methods also considered?

What measures were taken to ensure that data obtained from LC-MS/MS and other proteomic analyses are reproducible and not affected by technical variability?

Author Response

Answers for reviewer (1)

Thank you for your thoughtful suggestions and encouraging view. We appreciate the critical and positive comments. We have carefully considered each of the comments and revised the manuscript accordingly. Point-by-point responses to the comments are provided below, with your comments in black, our responses in blue, and the changes in the manuscript denoted in red fonts. The following are the list of the response.

The review presented by Kim et al. is interesting.

Can you provide more details on the size and diversity of the samples used for metabolomics and proteomics analysis? How do you account for the variability in cerebrospinal fluid (CSF) composition in different patients with Parkinson's disease?

à Thank you for your insightful comment. As our review is based on independent studies from the published literature, we did not classify into the sample size or diversity of the cohorts used in each metabolomics and proteomics study. However, we recognize that variability in CSF composition among PD patients is an important factor. To address this limitation, we have added a discussion in the manuscript acknowledging the potential variability across studies.

Line 368–375, Page 14

This review has several limitations: (i) The samples and analytical methods were not standardized, as we derived significantly changed metabolites and proteins from independent studies based on the existing literature. (ii) There is limited information regarding the mechanisms of microRNAs in PD, which precludes a comprehensive examination of single-omics and integrated-omics analyses. The combination of well-controlled patient samples, standardized analytical methods, and an accumulation of mechanistic studies involving microRNAs and other omics could substantially enhance our understanding of CSF changes in PD.

What steps have been taken to validate the biomarkers identified by metabolomics and proteomics in larger, independent cohorts?

à This review primarily synthesizes findings from existing literature and does not provide biomarkers identified by metabolomics and proteomics in larger, independent cohorts. We agree that the validation of these biomarkers in diverse populations is essential for their clinical applicability. The limitations of this review are added in the manuscript.

Line 368–375, Page 14

This review has several limitations: (i) The samples and analytical methods were not standardized, as we derived significantly changed metabolites and proteins from independent studies based on the existing literature. (ii) There is limited information regarding the mechanisms of microRNAs in PD, which precludes a comprehensive examination of single-omics and integrated-omics analyses. The combination of well-controlled patient samples, standardized analytical methods, and an accumulation of mechanistic studies involving microRNAs and other omics could substantially enhance our understanding of CSF changes in PD.

Considering that nuclear magnetic resonance (NMR) spectroscopy has a lower sensitivity compared to other methods such as mass spectrometry (MS), how did you ensure that all relevant metabolites were accurately detected?

à While it is true that NMR spectroscopy has lower sensitivity compared to MS, our review synthesizes findings from independent studies where metabolites were analyzed in both PD CSF and normal CSF using the same NMR method. This approach allows for a reliable statistical comparison of metabolite concentrations between the two groups, which enhances the validity of the findings. We believe that our collected data comes from published literature, which is strictly evaluated by peer reviewers in the scientific society.

How do you address the potential limitations of using in silico predictions in the study, particularly in predicting neurodegeneration and mitochondrial dysfunction?

à In our study, we recognize that there may be discrepancies between the changes observed in metabolites or proteins and the functional predictions made in silico. While we did not perform any artificial manipulations to address these discrepancies, we have highlighted these inconsistencies in our figures, where yellow lines indicate these inconsistencies automatically by the IPA (Ingenuity Pathway Analysis) software. This visual representation informs readers of the potential limitations and emphasizes the need for further investigation into these molecular relationships. In addition, this program is recognized as an internationally credible platform for integrated omics analysis, widely utilized by researchers for its robust and comprehensive capabilities in biological data interpretation.

<Figure 1B & Prediction Legend>

While the integration of metabolomics and proteomics data is valuable, how did you deal with potential discrepancies between these datasets when you constructed the integrated network for Parkinson's disease?

à While integrating metabolomics and proteomics data is indeed valuable, we agree that potential discrepancies between these datasets can arise. In our study, we did not manually adjust these discrepancies. Instead, we used an IPA program, which automatically handled data based on the manually curated knowledgebase and highlighted any inconsistencies in the integrated network. These areas of disagreement were indicated with yellow lines without further intervention.

<Figure 3B>

Could you explain the specific bioinformatics tools that were used to correlate the metabolomic and proteomic data and how these ensure reliable prediction of cellular functions and disease states?

à The reviewer suggested an excellent idea for in silico prediction and disease states. Our review used the Ingenuity Pathway Analysis (IPA) tool to analyze metabolomic and proteomic data. IPA offers several advantages, including its comprehensive, curated knowledgebase of biological functions, pathways, and interactions, which facilitates multi-omics data integration. By using IPA, we can reliably predict cellular functions and disease states through its advanced algorithms that evaluate the relevance of biomarkers in specific biological contexts. Furthermore, its extensive and continuously updated knowledge base, derived from peer-reviewed scientific literature, ensures that the tool stays relevant and reflective of the most current biological discoveries. We have added a discussion in the manuscript highlighting these advantages and how they enhance the reliability of our findings.

Line 174–178, Page 4

IPA is a powerful bioinformatics tool that enables the interpretation of complex omics data by identifying relevant biological pathways and networks, providing deep insights into disease mechanisms and therapeutic targets. Its extensive curated database makes it invaluable for transforming large datasets into interpretable biological knowledge.

The study predicts inhibition of mitochondrial functions and activation of neurodegeneration based on altered metabolite and protein levels. Can you provide experimental evidence that directly links these predictions to the observed pathological processes in PD?

à According to your comments, we added experimental information on the mitochondria and PD for the molecules emphasized in each metabolome (inosine and nitric oxide) and proteome (IL1B and NGF) based on published literatures.

Line 192–200, Page 4

Inosine, a type of purine, has been reported to increase mitochondrial respiration in cells under stressful conditions [63, 64], and inosine treatment has been demonstrated to have a neuroprotective function in models of PD [65, 66]. Additionally, a phase 3 clinical trial was conducted to treat PD through inosine intake (NCT02642393). Nitric oxide, despite its essential role in several signaling pathways, can cause oxidative stress and directly inhibit the mitochondrial respiratory chain [67, 68]. Several mechanisms by which nitric oxide contributes to neurotoxicity, particularly in PD, have been proposed, including axo-dendritic dysfunction, mitochondrial dysfunction, and disruptions in dopamine homeostasis [69, 70].

Line 240–250, Page 8

IL1B (interleukin 1 beta) is a proinflammatory cytokine involved in inflammatory responses from a wide range of stimuli, and its treatment has been reported to inhibit mitochondrial activity and function [99-101]. Koprich et al. confirmed that neuroinflammation can exacerbate the disease in a PD model, and that treatment with an IL1B receptor antagonist can reduce the loss of dopaminergic neurons [102]. Nerve growth factor (NGF) is a neurotrophin that plays a key role in various processes such as neuronal survival, differentiation, and maturation [103]. A decrease in NGF level has been observed in the serum of PD models [104, 105], and NGF infusion has been pursued in a single PD patient to support adrenal chromaffin tissue engrafted in the patient's putamen [106, 107]. The effects of NGF on mitochondrial function and dynamics support the multifaceted nature of NGF in relation to mitochondria and PD [108, 109].

Additional References

  1. Virag, L.; Szabo, C. Purines inhibit poly(ADP-ribose) polymerase activation and modulate oxidant-induced cell death. FASEB J. 2001, 15, 99-107.
  2. Li, M. X.; Wu, X. T.; Jing, W. Q.; Hou, W. K.; Hu, S.; Yan, W. Inosine enhances tumor mitochondrial respiration by inducing Rag GTPases and nascent protein synthesis under nutrient starvation. Cell Death Dis. 2023, 14, 492.
  3. El-Latif, A. M. A.; Rabie, M. A.; Sayed, R. H.; Fattah, M.; Kenawy, S. A. Inosine attenuates rotenone-induced Parkinson's disease in rats by alleviating the imbalance between autophagy and apoptosis. Drug Dev. Res. 2023, 84, 1159-1174.
  4. Khanal, S.; Bok, E.; Kim, J.; Park, G. H.; Choi, D. Y. Dopaminergic neuroprotective effects of inosine in MPTP-induced parkinsonian mice via brain-derived neurotrophic factor upregulation. Neuropharmacology 2023, 238, 109652.
  5. Brown, G. C. Nitric oxide and mitochondria. Front. Biosci. 2007, 12, 1024-1033.
  6. Tengan, C. H.; Moraes, C. T. NO control of mitochondrial function in normal and transformed cells. Biochim. Biophys. Acta Bioenerg. 2017, 1858, 573-581.
  7. Zhang, L.; Dawson, V. L.; Dawson, T. M. Role of nitric oxide in Parkinson's disease. Pharmacol. Ther. 2006, 109, 33-41.
  8. Stykel, M. G.; Ryan, S. D. Nitrosative stress in Parkinson's disease. NPJ Parkinsons Dis. 2022, 8, 104.

  1. Verma, G.; Bhatia, H.; Datta, M. JNK1/2 regulates ER-mitochondrial Ca2+ cross-talk during IL-1beta-mediated cell death in RINm5F and human primary beta-cells. Mol. Biol. Cell 2013, 24, 2058-2071.
  2. Lopez-Armada, M. J.; Carames, B.; Martin, M. A.; Cillero-Pastor, B.; Lires-Dean, M.; Fuentes-Boquete, I.; Arenas, J.; Blanco, F. J. Mitochondrial activity is modulated by TNFalpha and IL-1beta in normal human chondrocyte cells. Osteoarthritis Cartilage 2006, 14, 1011-1022.
  3. Aarreberg, L. D.; Esser-Nobis, K.; Driscoll, C.; Shuvarikov, A.; Roby, J. A.; Gale, M., Jr. Interleukin-1beta Induces mtDNA Release to Activate Innate Immune Signaling via cGAS-STING. Mol. Cell 2019, 74, 801-815.e6.
  4. Koprich, J. B.; Reske-Nielsen, C.; Mithal, P.; Isacson, O. Neuroinflammation mediated by IL-1beta increases susceptibility of dopamine neurons to degeneration in an animal model of Parkinson's disease. J. Neuroinflammation 2008, 5, 8.
  5. Freed, W. J. The role of nerve-growth factor (NGF) in the central nervous system. Brain Res. Bull. 1976, 1, 393-412.
  6. Lorigados, L.; Alvarez, P.; Pavon, N.; Serrano, T.; Blanco, L.; Macias, R. NGF in experimental models of Parkinson disease. Mol. Chem. Neuropathol. 1996, 28, 225-228.
  7. Lorigados Pedre, L.; Pavon Fuentes, N.; Alvarez Gonzalez, L.; McRae, A.; Serrano Sanchez, T.; Blanco Lescano, L.; Macias Gonzalez, R. Nerve growth factor levels in Parkinson disease and experimental parkinsonian rats. Brain Res. 2002, 952, 122-127.
  8. Aloe, L.; Rocco, M. L.; Bianchi, P.; Manni, L. Nerve growth factor: from the early discoveries to the potential clinical use. J. Transl. Med. 2012, 10, 239.
  9. Olson, L.; Backlund, E. O.; Ebendal, T.; Freedman, R.; Hamberger, B.; Hansson, P.; Hoffer, B.; Lindblom, U.; Meyerson, B.; Stromberg, I., et al. Intraputaminal infusion of nerve growth factor to support adrenal medullary autografts in Parkinson's disease. One-year follow-up of first clinical trial. Arch. Neurol. 1991, 48, 373-381.
  10. Chada, S. R.; Hollenbeck, P. J. Nerve growth factor signaling regulates motility and docking of axonal mitochondria. Curr. Biol. 2004, 14, 1272-1276.
  11. Martorana, F.; Gaglio, D.; Bianco, M. R.; Aprea, F.; Virtuoso, A.; Bonanomi, M.; Alberghina, L.; Papa, M.; Colangelo, A. M. Differentiation by nerve growth factor (NGF) involves mechanisms of crosstalk between energy homeostasis and mitochondrial remodeling. Cell Death Dis. 2018, 9, 391.

What impact do the observed changes in specific metabolites (such as inosine, levodopa and nitric oxide) have on the clinical course of Parkinson's disease?

à As you have pointed out, we have included additional information regarding the clinical implications of the metabolites (inosine and nitric oxide) highlighted in the manuscript.

Line 192–200, Page 4

Inosine, a type of purine, has been reported to increase mitochondrial respiration in cells under stressful conditions [63, 64], and inosine treatment has been demonstrated to have a neuroprotective function in models of PD [65, 66]. Additionally, a phase 3 clinical trial was conducted to treat PD through inosine intake (NCT02642393). Nitric oxide, despite its essential role in several signaling pathways, can cause oxidative stress and directly inhibit the mitochondrial respiratory chain [67, 68]. Several mechanisms by which nitric oxide contributes to neurotoxicity, particularly in PD, have been proposed, including axo-dendritic dysfunction, mitochondrial dysfunction, and disruptions in dopamine homeostasis [69, 70].

Additional References

  1. Virag, L.; Szabo, C. Purines inhibit poly(ADP-ribose) polymerase activation and modulate oxidant-induced cell death. FASEB J. 2001, 15, 99-107.
  2. Li, M. X.; Wu, X. T.; Jing, W. Q.; Hou, W. K.; Hu, S.; Yan, W. Inosine enhances tumor mitochondrial respiration by inducing Rag GTPases and nascent protein synthesis under nutrient starvation. Cell Death Dis. 2023, 14, 492.
  3. El-Latif, A. M. A.; Rabie, M. A.; Sayed, R. H.; Fattah, M.; Kenawy, S. A. Inosine attenuates rotenone-induced Parkinson's disease in rats by alleviating the imbalance between autophagy and apoptosis. Drug Dev. Res. 2023, 84, 1159-1174.
  4. Khanal, S.; Bok, E.; Kim, J.; Park, G. H.; Choi, D. Y. Dopaminergic neuroprotective effects of inosine in MPTP-induced parkinsonian mice via brain-derived neurotrophic factor upregulation. Neuropharmacology 2023, 238, 109652.
  5. Brown, G. C. Nitric oxide and mitochondria. Front. Biosci. 2007, 12, 1024-1033.
  6. Tengan, C. H.; Moraes, C. T. NO control of mitochondrial function in normal and transformed cells. Biochim. Biophys. Acta Bioenerg. 2017, 1858, 573-581.
  7. Zhang, L.; Dawson, V. L.; Dawson, T. M. Role of nitric oxide in Parkinson's disease. Pharmacol. Ther. 2006, 109, 33-41.
  8. Stykel, M. G.; Ryan, S. D. Nitrosative stress in Parkinson's disease. NPJ Parkinsons Dis. 2022, 8, 104.

What are the next steps to validate the identified biomarkers for clinical use and how do you plan to address potential challenges in moving from research findings to practical diagnostic tools?

à Analysis of large-scale data through machine learning can lead to deriving more meaningful biomarkers from the CSF of PD patients and clinically applicable patterns of metabolite and proteome changes. A brief description of this has been added to the text.

Line 360–362, Page 14

Furthermore, by analyzing large datasets, ML can uncover subtle patterns and aid in identifying biomarker candidates for PD by detecting of omics-based alteration patterns, thereby advancing the development of reliable biomarkers.

Can you explain any known limitations or challenges in using CSF-based biomarkers to differentiate between Parkinson's disease and other neurodegenerative diseases, such as multiple system atrophy (MSA)?

à Several limitations and challenges are associated with using CSF-based biomarkers to differentiate between PD and other neurodegenerative diseases, such as multiple system atrophy (MSA). These include the sensitivity limitations of target-based analytical methods, challenges in multi-dimensional bioinformatic analysis, and the need for advanced technologies to detect low-abundance biomarkers. In the manuscript, we added the limitations and challenges related to these.

Line 120–136, Page 3

Nowadays, there are limitations in using CSF-based biomarkers to differentiate between PD and other neurodegenerative diseases, such as MSA. Most analytical approaches for detecting metabolites in CSF are target-based, using NMR and GC-MS. These methods also face challenges, as molecules below nanomolar concentrations can hardly be detected. Moreover, there are significant constraints in the multi-dimensional bioinformatic analysis of proteins and metabolites, and the related software has not yet been fully developed. To overcome these limitations, non-targeted approaches are being employed, which allow for the detection of a broader range of novel molecules. Advanced technologies, such as nanopore sensing and surface-enhanced Raman spectroscopy [43, 44], which can detect molecules at picomolar and femtomolar concentrations, are also being integrated into these analyses. Finally, as large datasets are obtained, a new challenge arises in extracting multidimensional molecular data through dimensional reduction and analyzing it using various artificial intelligence (AI) algorithms. These analyses will be based on networks that accurately reflect neurodegenerative disease-specific molecular pathways, facilitating the identification of biomarkers to differentiate between PD and other neurodegenerative diseases. This is briefly discussed in section 5: Integration of Metabolomics and Proteomics in CSF of Patients with PD.

Additional References

  1. Jeong, K. B.; Ryu, M.; Kim, J. S.; Kim, M.; Yoo, J.; Chung, M.; Oh, S.; Jo, G.; Lee, S. G.; Kim, H. M., et al. Single-molecule fingerprinting of protein-drug interaction using a funneled biological nanopore. Nat. Commun. 2023, 14, 1461.
  2. Shin, H.; Oh, S.; Kang, D.; Choi, Y. Protein Quantification and Imaging by Surface-Enhanced Raman Spectroscopy and Similarity Analysis. Adv. Sci. (Weinh.) 2020, 7, 1903638.

Given the advances in analytical techniques, why were certain methods such as gas chromatography-mass spectrometry (GC-MS) chosen over others for metabolome profiling? Were other methods also considered?

à In metabolome data, many metabolites might be analyzed using GC-MS due to its robust analytical capabilities and well-established methodology for a wide range of metabolites, including organic acids, fatty acids, lipids, and sugars. GC-MS is particularly effective for metabolomic studies as it provides high sensitivity and specificity, which are essential for accurate metabolite identification and quantification. We do not have specific information on why particular analytical methods were used for individual metabolites, but GC-MS was likely chosen due to its many advantages.

             While researchers recognize the strengths of GC-MS, they also considered other analytical techniques, including liquid chromatography with electrochemical detection (LC-ECD), liquid chromatography-tandem mass spectrometry (LC-MS/MS), isotope dilution mass spectrometry (IDMS), and nuclear magnetic resonance (NMR) spectroscopy.

What measures were taken to ensure that data obtained from LC-MS/MS and other proteomic analyses are reproducible and not affected by technical variability?

à Due to the nature of our review, which synthesizes findings from independent studies, we did not conduct original experiments to establish the reproducibility of data obtained from LC-MS/MS and other proteomic analyses. We acknowledge that variations in technical procedures across different studies can introduce variability in the results. To address this limitation, we have included a discussion in the manuscript highlighting the importance of standardization in methodologies and the need for future research to ensure data reproducibility.

Line 368–375, Page 14

This review has several limitations: (i) The samples and analytical methods were not standardized, as we derived significantly changed metabolites and proteins from independent studies based on the existing literature. (ii) There is limited information regarding the mechanisms of microRNAs in PD, which precludes a comprehensive examination of single-omics and integrated-omics analyses. The combination of well-controlled patient samples, standardized analytical methods, and an accumulation of mechanistic studies involving microRNAs and other omics could substantially enhance our understanding of CSF changes in PD.

Reviewer 2 Report

Comments and Suggestions for Authors

The authors try to perform integrative metabolome and proteome analysis of CSF in PD, which will be significant to new biomarker and therapeutic targets identification and development. 

Although the article is interesting, some concerns are raised:

1, the authors have brief discussion on microRNA in CSF of PD patients. The authors should summarize recent findings on microRNA CSF biomarkers and show them in a new table. If integrative study can be performed on metabolome, proteome and microRNA profiles in CSF, more interesting findings should be achieved. This is because some protein can be the target of microRNA. The uncovered linkage among them will be more significant. 

2, the authors should highlight the new findings from integrative analysis from metabolome and proteome. New biomarker and pathways should be highlighted and discussed in more detailed contents. 

Author Response

Answers for reviewer (2)

We sincerely thank you for your thoughtful and valuable suggestions. We appreciate the constructive comments. Each comment has been carefully reviewed, and we have revised the manuscript accordingly. Below, we would like to provide point-by-point responses to your comments. The original comments are displayed in black, our responses are in blue, and the changes in the manuscript are indicated in red text.

The authors try to perform integrative metabolome and proteome analysis of CSF in PD, which will be significant to new biomarker and therapeutic targets identification and development.

Although the article is interesting, some concerns are raised:

1, the authors have brief discussion on microRNA in CSF of PD patients. The authors should summarize recent findings on microRNA CSF biomarkers and show them in a new table. If integrative study can be performed on metabolome, proteome and microRNA profiles in CSF, more interesting findings should be achieved. This is because some protein can be the target of microRNA. The uncovered linkage among them will be more significant.

à As you have advised, we also agree that microRNAs in the CSF of PD patients must be considered. However, although many studies have analyzed microRNAs in the CSF of PD patients, single-omics analyses of microRNAs have been limited due to the lack of information on their biological functions or relationship with diseases in the IPA program. In addition, we thought that a detailed discussion of microRNAs was beyond the scope of our review, as our primary focus is on proteins and metabolites outside of cells, from CSF. Since microRNAs primarily function in transcriptional regulation within cells, their role is distant from our focus on extracellular proteomes and metabolomes in cell-free CSF.

             However, as we believe that microRNA analysis remains important, we have provided a more detailed description of several studies analyzing microRNA in the CSF of PD patients and tabulated PD-related microRNA information available from IPA (Table S5). We also described the limitations of microRNAs not covered in this review and emphasized the importance of their analysis.

Table S5. Ingenuity Pathway Analysis (IPA)-based Parkinson’s disease-related microRNAs.

Symbol

Synonyms

Increased/
decreased

Source

Reference

miR-126a-5p

hsa-miR-126, hsa-miR-126-5p, miR-126, miR-126-5p, miR-126a-5p

Decreased

PBMC

[143]

miR-151-3p

hsa-miR-151-3p, hsa-miR-151a-3p, miR-151, miR-151-3p, miR-151a-3p

Decreased

PBMC

miR-151-5p

hsa-miR-151-5p, hsa-miR-151a-5p, hsa-miR-151b, miR-151, miR-151-5p, miR-151a-5p, miR-151b

Decreased

PBMC

miR-199a-5p

hsa-miR-199-s, hsa-miR-199a, hsa-mir-199a-1-5p, hsa-mir-199a-2-5p, hsa-miR-199a-5p, hsa-miR-199b-5p, miR-199a-5p, miR-199b, miR-199b-5p, Mir199-5p

Decreased

PBMC

miR-23a-3p

hsa-miR-23a-3p, hsa-miR-23b-3p, hsa-miR-23c, miR-23a, miR-23a-3p, miR-23b-3p, miR-23c

Increased

Plasma

[144]

Decreased

Small extracellular vesicles from plasma

miR-708-5p

hsa-miR-3139, hsa-miR-28-5p, hsa-miR-708-5p, miR-3139, miR-28-5p, miR-708-5p, MIR28A-5p

Decreased

PBMC

[143]

mir-126

HSA-MIR-123, hsa-miR-126, MI0000149, microRNA 123, microRNA 126, microRNA 126a, microRNA 126b, Mir3567, miR-123, Mir126a, Mir126b, MIRN123, MIRN126, miRNA126

Decreased

PBMC

mir-130

HSA-MIR-130, hsa-miR-301, hsa-miR-130a, hsa-miR-130b, hsa-miR-301a, hsa-miR-301b, microRNA 130, microRNA 301, microRNA 130a, microRNA 130b, microRNA 301a, microRNA 301b, MIR301, MIR130A, MIR130B, MIR301A, MIR301B, Mirn130, MIRN301, MIRN130A, MIRN130B, MIRN301A, MIRN301B, miRNA130A

Decreased

PBMC

mir-147

hsa-mir-147, hsa-mir-147a, hsa-miR-147b, microRNA 147, microRNA 147a, microRNA 147b, MIR147A, MIR147B, MIRN147, MIRN147B

Decreased

PBMC

mir-19

C13orf25, HSA-MIR-19A, hsa-miR-19b, hsa-mir-19b-1, hsa-mir-19b-2, microRNA 19, microRNA 19a, microRNA 19b, microRNA 19b-1, microRNA 19b-2, MIR17HG, MIR19A, MIR19A., miR-19a/b, MIR19B, MIR19B1, MIR19B2, MIRH1, MIRHG1, MIRN19A, Mirn19b, MIRN19B1, MIRN19B2, miRNA19A, miRNA19B1

Decreased

PBMC

mir-199

HSA-MIR-199A, hsa-mir-199a-1, hsa-mir-199a-2, Hsa-mir199a-as, hsa-miR-199b, hsa-miR-199-s, MI0000280, microRNA 199, microRNA 199a-1, microRNA 199a-2, microRNA 199b, mir199 3p, mir199 5p, MIR199A, MIR199A1, MIR199A2, miR-199a-3p(2), miR-199a-5p(2), MIR199B, MIR-199-s, Mirn199, Mirn199a, MIRN199A1, MIRN199A2, Mirn199a-as, MIRN199B

Decreased

PBMC

mir-23

hsa-miR-23a, hsa-miR-23a/b, hsa-miR-23b, microRNA 23a, microRNA 23b, MIR23A, MIR23B, MIRN23A, MIRN23B, miRNA23A, miRNA23B

Increased

Plasma

[144]

mir-26

HSA-MIR-26A, hsa-miR-26a-1, hsa-mir-26a-2, hsa-miR-26b, MI0000574, microRNA 26a, microRNA 26a-1, microRNA 26a-2, microRNA 26b, MIR26A1, MIR26A2, MIR26A, MIR26B, MIRN26A1, MIRN26A2, MIRN26A, MIRN26B

Decreased

PBMC

[143]

mir-28

hsa-miR-28, hsa-miR-151, hsa-mir-151a, hsa-mir-151b, microRNA 28, microRNA 151, microRNA 151a, microRNA 151b, microRNA 28A, microRNA 28c, MIR151, Mir3586, miR-28-A, MIR151A, MIR151B, Mir28c, MIRN28, MIRN151

Decreased

PBMC

mir-29

hsa-miR-102, hsa-mir-29, hsa-miR-29a, HSA-MIR-29B, hsa-mir-29b-1, hsa-mir-29b-2, hsa-mir-29b-3, hsa-miR-29c, MI0000106, microRNA 29, microRNA 29a, microRNA 29b, microRNA 29b-1, microRNA 29b-2, microRNA 29c, microRNA 29 group, microRNA mir-29b-3, microRNA mir-29c-2, miR-102, MIR29A, MIR29B, MIR29B1, MIR29B2, Mir29b-3, MIR29C, Mir29c-2, miR-29 family, MIRN29, MIRN29A, MIRN29B, MIRN29B1, MIRN29B2, MIRN29C, miRNA29A, miRNA29B1, miRNA29C

Decreased

PBMC

mir-30

hsa-miR-30, hsa-miR-30a, hsa-miR-30a-3p, hsa-miR-30b, HSA-MIR-30BN, HSA-MIR-30C, hsa-mir-30c-1, hsa-mir-30c-2, HSA-MIR-30D, hsa-miR-30e, hsa-miR-30e-5p, hsa-mir-97-6, MI0000099, microRNA 30a, microRNA 30B, microRNA 30C, microRNA 30c-1, microRNA 30c-2, microRNA 30d, microRNA 30e, miR-200, MIR30A, MIR30A3P, MIR30B, MIR30C, MIR30C1, MIR30C2, MIR30D, miR-30d-prec, MIR30E, Mir97, MIRN30A, MIRN30B, Mirn30c, MIRN30C1, MIRN30C2, MIRN30D, MIRN30E, MIRN30E-5P

Decreased

PBMC

mir-335

hsa-miR-335, microRNA 335, microRNA mir-335, MIRN335, miRNA335

Decreased

PBMC

mir-374

hsa-miR-374, hsa-miR-374a, hsa-miR-374b, hsa-miR-374c, microRNA 374, microRNA 374a, microRNA 374b, microRNA 374c, MIR374A, MIR374B, MIR374C, MIRN374, MIRN374A, MIRN374B`

Decreased

PBMC

*PBMC: peripheral blood mononuclear cell

Line 329–336, Page 13–14

For example, Tan et al. found 21 up-regulated microRNAs in the early stage of PD and suggested miR-409-3p as an indicator of early PD [141]. Burgos et al. found 17 microRNAs differentially expressed in the CSF of PD patients, highlighting five microRNAs (miR-127-3p, miR-443, miR-431-3p, miR-136-3p, and miR-10a-5p) commonly differentially expressed in both PD and Alzheimer’s disease compared to control subjects [142]. Additionally, we obtained a list of microRNAs associated with PD from the IPA database [143, 144] (Table S5). Although efforts to find microRNA markers in CSF are ongoing, the underlying mechanisms of microRNAs in PD remain poorly understood.

Line 368–375, Page 14

This review has several limitations: (i) The samples and analytical methods were not standardized, as we derived significantly changed metabolites and proteins from independent studies based on the existing literature. (ii) There is limited information regarding the mechanisms of microRNAs in PD, which precludes a comprehensive examination of single-omics and integrated-omics analyses. The combination of well-controlled patient samples, standardized analytical methods, and an accumulation of mechanistic studies involving microRNAs and other omics could substantially enhance our understanding of CSF changes in PD.

Additional References

  1. Tan, X.; Hu, J.; Ming, F.; Lv, L.; Yan, W.; Peng, X.; Bai, R.; Xiao, Q.; Zhang, H.; Tang, B., et al. MicroRNA-409-3p Targeting at ATXN3 Reduces the Apoptosis of Dopamine Neurons Based on the Profile of miRNAs in the Cerebrospinal Fluid of Early Parkinson's Disease. Front. Cell Dev. Biol. 2021, 9, 755254.
  2. Burgos, K.; Malenica, I.; Metpally, R.; Courtright, A.; Rakela, B.; Beach, T.; Shill, H.; Adler, C.; Sabbagh, M.; Villa, S., et al. Profiles of extracellular miRNA in cerebrospinal fluid and serum from patients with Alzheimer's and Parkinson's diseases correlate with disease status and features of pathology. PLoS One 2014, 9, e94839.
  3. Martins, M.; Rosa, A.; Guedes, L. C.; Fonseca, B. V.; Gotovac, K.; Violante, S.; Mestre, T.; Coelho, M.; Rosa, M. M.; Martin, E. R., et al. Convergence of miRNA expression profiling, alpha-synuclein interacton and GWAS in Parkinson's disease. PLoS One 2011, 6, e25443.
  4. Rai, S.; Bharti, P. S.; Singh, R.; Rastogi, S.; Rani, K.; Sharma, V.; Gorai, P. K.; Rani, N.; Verma, B. K.; Reddy, T. J., et al. Circulating plasma miR-23b-3p as a biomarker target for idiopathic Parkinson's disease: comparison with small extracellular vesicle miRNA. Front. Neurosci. 2023, 17, 1174951.

2, the authors should highlight the new findings from integrative analysis from metabolome and proteome. New biomarker and pathways should be highlighted and discussed in more detailed contents.

à As per your comments, to highlight the biomarkers, we have included detailed information regarding the relationship between mitochondria and PD for the molecules emphasized in each metabolome and proteome. In addition, we described the relationships among the molecules highlighted in the integrated omics. We also created a table that provides comprehensive information on the canonical pathway analysis, as well as the biological functions and diseases associated with the integrated omics (Table S14).

Table S1. List of the top seven inhibited canonical pathways identified through integrative omics analysis.

No.

Ingenuity canonical pathways

−log(p-value)

Activation
z-score

Related molecules

1

Neutrophil degranulation

2.96

-2.121

CO3, CD44, CH3L1, GROA, SAP3, HPT, PTPR2, TTHY

2

Neurotransmitter clearance

4.97

-2.000

3,4-dihydroxyphenylacetic acid, 5-hydroxyindol-3-acetic acid, 5-hydroxytryptamine, homovanillic acid

3

UFMylation signaling

3.66

-2.000

IL1B, IL6, L-glutamic acid, TNF

4

Sulfur amino acid metabolism

3.57

-2.000

betaine, glutathione disulfide, L-glutamic acid, taurine

5

Cargo recognition for clathrin-mediated endocytosis

2.77

-2.000

AP2B1, APOB, TGFA, TGON2

6

Clathrin-mediated endocytosis

2.44

-2.000

AP2B1, APOB, TGFA, TGON2

7

IL-10 signaling

2.28

-2.000

IL1B, IL6, nitric oxide, TNF

Table S2. List of the top seven activated canonical pathways identified through integrative omics analysis.

No.

Ingenuity canonical pathways

−log(p-value)

Activation
z-score

Related molecules

1

Communication between innate and adaptive immune cells

1.31

2.828

IGHG4, HV102, KV401, LV147, IL1B, IL4, IL6, TNF

2

IL-33 signaling

3.77

2.449

CCL2, IL1B, IL4, IL6, nitric oxide, TNF

3

Activin inhibin signaling

2.63

2.236

IGHG4, IL1B, IL6, TGFB1, TNF

4

FCGR dependent phagocytosis

3.04

2.236

arachidonic acid, IGHG4, HV102, KV401, LV147

5

MAPK signaling
(inhibiting influenza)

4.54

2.236

arachidonic acid, CCL2, IL1B, IL6, TNF

6

NAFLD signaling

4.99

2.121

APOB, D-fructose, IL17A, IL1B, IL4, IL6, TGFB1, TNF

7

Dendritic cell maturation

0.59

2.000

IGHG4, IL1B, IL6, TNF

Table S3. List of the top seven inhibited diseases and biological functions identified through integrative omics analysis.

No.

Diseases or functions annotation

−log(p-value)

Activation
z-score

Related molecules

1

Tubulation of vascular endothelial cells

8.86

-2.894

5-hydroxytryptamine, APOH, CD44, GROA, SCF, SCG2, PEDF, TGFB1, TNF, VEGFA

2

Respiration of mitochondria

7.28

-2.63

betaine, inosine, L-glutamic acid, nitric oxide, spermidine, TGFB1, TNF

3

Mobilization of Ca2+

11.91

-2.579

5-hydroxytryptamine, CO3, CO4A/B, CCL2, CCL23, CCL28, CCL8, CD44, corticosterone, GROA, IL16, IL4, SCF, L-glutamic acid, NGF, NPY, TGFB1, TNF, VEGFA

4

Release of neurotransmitter

9.15

-2.578

2-phenethylamine, 5-hydroxytryptamine, CO3, corticosterone, IL1B, L-glutamic acid, levodopa, NGF, nitric oxide, NPY, PARK7, TNF

5

Proliferation of neuronal cells

9.67

-2.389

5-hydroxytryptamine, APOE, CCL2, CLUS, CSF1, ENPP2, IL17A, IL1B, IL6, inosine, SCF, levodopa, NGF, nitric oxide, NPY, putrescine, TGFA, TGFB1, TNF,VEGFA,VGF

6

Transmembrane potential of mitochondria

6.48

-2.354

arachidonic acid, IL1B, IL4, IL6, L-glutamic acid, levodopa, N-methyl-(R)-salsolinol, NGF, nitric oxide, PARK7, TGFB1, TNF

7

Growth of epithelial tissue

10.97

-2.894

5-hydroxytryptamine, APOH, CD44, GROA, SCF, SCG2, PEDF, TGFB1, TNF, VEGFA

Table S4. List of the top seven activated diseases and biological functions identified through integrative omics analysis.

No.

Diseases or functions annotation

−log(p-value)

Activation
z-score

Related molecules

1

Apoptosis of vascular cells

8.95

3.147

5-hydroxytryptamine, arachidonic acid, PD1L1, IL1B, IL6, SCF, RET4, SCG2, taurine, TGFB1, TNF, VEGFA

2

Apoptosis of vascular endothelial cells

7.12

2.72

5-hydroxytryptamine, arachidonic acid, PD1L1, IL1B, RET4, SCG2, TGFB1, TNF, VEGFA

3

Visceromegaly

7.50

2.586

5-hydroxytryptamine, APOE, CO3, CO4A/B, CMGA, corticosterone, CSF1, HPT, IL17A, IL1B, IL4, IL6, nitric oxide, PARK7, NEC1, putrescine, RET4, 7B2, spermidine, taurine, TGFB1, TNF, VEGFA

4

Adhesion of connective tissue cells

7.28

2.449

CCL2, CD44, corticosterone, CSF1, IL1B, IL4, IL6, TGFB1, TNF, VEGFA

5

Aggregation of phagocytes

10.08

2.384

CD44, IL1B, IL4, IL6, SCF, KLKB1, TGFB1, TNF

6

Adhesion of lymphocytes

7.39

2.377

arachidonic acid, CCL2, CCL28, CD44, IL1B, IL4, IL6, TGFB1, TNF

7

Abnormal morphology of body cavity

8.15

2.356

5-hydroxytryptamine, APOE, CO3, CO4A/B, CCL28, PD1L1, CD44, CMGA, CLUS, corticosterone, CSF1, DNS2A, FGF19, HPT, SIAL, IL17A, IL1B, IL4, IL6, nitric oxide, PARK7, NEC1, PTPR2, putrescine, RET4, 7B2, spermidine, taurine, TGFA, TGFB1, TNF, VEGFA, VGF

Line 192–200, Page 4

Inosine, a type of purine, has been reported to increase mitochondrial respiration in cells under stressful conditions [63, 64], and inosine treatment has been demonstrated to have a neuroprotective function in models of PD [65, 66]. Additionally, a phase 3 clinical trial was conducted to treat PD through inosine intake (NCT02642393). Nitric oxide, despite its essential role in several signaling pathways, can cause oxidative stress and directly inhibit the mitochondrial respiratory chain [67, 68]. Several mechanisms by which nitric oxide contributes to neurotoxicity, particularly in PD, have been proposed, including axo-dendritic dysfunction, mitochondrial dysfunction, and disruptions in dopamine homeostasis [69, 70].

Line 240–250, Page 8

IL1B (interleukin 1 beta) is a proinflammatory cytokine involved in inflammatory responses from a wide range of stimuli, and its treatment has been reported to inhibit mitochondrial activity and function [99-101]. Koprich et al. confirmed that neuroinflammation can exacerbate the disease in a PD model, and that treatment with an IL1B receptor antagonist can reduce the loss of dopaminergic neurons [102]. Nerve growth factor (NGF) is a neurotrophin that plays a key role in various processes such as neuronal survival, differentiation, and maturation [103]. A decrease in NGF level has been observed in the serum of PD models [104, 105], and NGF infusion has been pursued in a single PD patient to support adrenal chromaffin tissue engrafted in the patient's putamen [106, 107]. The effects of NGF on mitochondrial function and dynamics support the multifaceted nature of NGF in relation to mitochondria and PD [108, 109].

Line 286–292, Page 12

The integrated network underscores the connectivity among nitric oxide, IL1B, and NGF among the molecules highlighted in the single omics analyses. The proportional relationship between IL1B and nitric oxide has been proven through studies demonstrating that nitric oxide production is induced by IL1B treatment or that both IL1B and nitric oxide levels increase concurrently in response to an inflammatory response [129, 130]. In contrast, NGF and nitric oxide have an inverse proportion, as supported by the research indicating that nitric oxide suppresses NGF production [131].

Line 299–306, Page 12

In particular, various inflammatory and immune-related pathways are predicted to be activated, with molecules such as IL1B, IL4, IL6, TNF, and nitric oxide appearing to play significant roles (Table S1, Table S2). These observations reflect the activation of multiple immune response pathways in PD. Among the top seven activated/inhibited diseases and biological functions, vascular endothelial cell damage and immune cell responses were predicted to be activated. Conversely, mitochondrial and neuronal cell functions were expected to be inhibited in patients with PD (Figure 3D, Table S3, and Table S4).

Additional References

  1. Virag, L.; Szabo, C. Purines inhibit poly(ADP-ribose) polymerase activation and modulate oxidant-induced cell death. FASEB J. 2001, 15, 99-107.
  2. Li, M. X.; Wu, X. T.; Jing, W. Q.; Hou, W. K.; Hu, S.; Yan, W. Inosine enhances tumor mitochondrial respiration by inducing Rag GTPases and nascent protein synthesis under nutrient starvation. Cell Death Dis. 2023, 14, 492.
  3. El-Latif, A. M. A.; Rabie, M. A.; Sayed, R. H.; Fattah, M.; Kenawy, S. A. Inosine attenuates rotenone-induced Parkinson's disease in rats by alleviating the imbalance between autophagy and apoptosis. Drug Dev. Res. 2023, 84, 1159-1174.
  4. Khanal, S.; Bok, E.; Kim, J.; Park, G. H.; Choi, D. Y. Dopaminergic neuroprotective effects of inosine in MPTP-induced parkinsonian mice via brain-derived neurotrophic factor upregulation. Neuropharmacology 2023, 238, 109652.
  5. Brown, G. C. Nitric oxide and mitochondria. Front. Biosci. 2007, 12, 1024-1033.
  6. Tengan, C. H.; Moraes, C. T. NO control of mitochondrial function in normal and transformed cells. Biochim. Biophys. Acta Bioenerg. 2017, 1858, 573-581.
  7. Zhang, L.; Dawson, V. L.; Dawson, T. M. Role of nitric oxide in Parkinson's disease. Pharmacol. Ther. 2006, 109, 33-41.
  8. Stykel, M. G.; Ryan, S. D. Nitrosative stress in Parkinson's disease. NPJ Parkinsons Dis. 2022, 8, 104.

  1. Verma, G.; Bhatia, H.; Datta, M. JNK1/2 regulates ER-mitochondrial Ca2+ cross-talk during IL-1beta-mediated cell death in RINm5F and human primary beta-cells. Mol. Biol. Cell 2013, 24, 2058-2071.
  2. Lopez-Armada, M. J.; Carames, B.; Martin, M. A.; Cillero-Pastor, B.; Lires-Dean, M.; Fuentes-Boquete, I.; Arenas, J.; Blanco, F. J. Mitochondrial activity is modulated by TNFalpha and IL-1beta in normal human chondrocyte cells. Osteoarthritis Cartilage 2006, 14, 1011-1022.
  3. Aarreberg, L. D.; Esser-Nobis, K.; Driscoll, C.; Shuvarikov, A.; Roby, J. A.; Gale, M., Jr. Interleukin-1beta Induces mtDNA Release to Activate Innate Immune Signaling via cGAS-STING. Mol. Cell 2019, 74, 801-815.e6.
  4. Koprich, J. B.; Reske-Nielsen, C.; Mithal, P.; Isacson, O. Neuroinflammation mediated by IL-1beta increases susceptibility of dopamine neurons to degeneration in an animal model of Parkinson's disease. J. Neuroinflammation 2008, 5, 8.
  5. Freed, W. J. The role of nerve-growth factor (NGF) in the central nervous system. Brain Res. Bull. 1976, 1, 393-412.
  6. Lorigados, L.; Alvarez, P.; Pavon, N.; Serrano, T.; Blanco, L.; Macias, R. NGF in experimental models of Parkinson disease. Mol. Chem. Neuropathol. 1996, 28, 225-228.
  7. Lorigados Pedre, L.; Pavon Fuentes, N.; Alvarez Gonzalez, L.; McRae, A.; Serrano Sanchez, T.; Blanco Lescano, L.; Macias Gonzalez, R. Nerve growth factor levels in Parkinson disease and experimental parkinsonian rats. Brain Res. 2002, 952, 122-127.
  8. Aloe, L.; Rocco, M. L.; Bianchi, P.; Manni, L. Nerve growth factor: from the early discoveries to the potential clinical use. J. Transl. Med. 2012, 10, 239.
  9. Olson, L.; Backlund, E. O.; Ebendal, T.; Freedman, R.; Hamberger, B.; Hansson, P.; Hoffer, B.; Lindblom, U.; Meyerson, B.; Stromberg, I., et al. Intraputaminal infusion of nerve growth factor to support adrenal medullary autografts in Parkinson's disease. One-year follow-up of first clinical trial. Arch. Neurol. 1991, 48, 373-381.
  10. Chada, S. R.; Hollenbeck, P. J. Nerve growth factor signaling regulates motility and docking of axonal mitochondria. Curr. Biol. 2004, 14, 1272-1276.
  11. Martorana, F.; Gaglio, D.; Bianco, M. R.; Aprea, F.; Virtuoso, A.; Bonanomi, M.; Alberghina, L.; Papa, M.; Colangelo, A. M. Differentiation by nerve growth factor (NGF) involves mechanisms of crosstalk between energy homeostasis and mitochondrial remodeling. Cell Death Dis. 2018, 9, 391.

  1. Eitner, A.; Muller, S.; Konig, C.; Wilharm, A.; Raab, R.; Hofmann, G. O.; Kamradt, T.; Schaible, H. G. Inhibition of Inducible Nitric Oxide Synthase Prevents IL-1beta-Induced Mitochondrial Dysfunction in Human Chondrocytes. Int. J. Mol. Sci. 2021, 22, 2477.
  2. Sudo, K.; Takezawa, Y.; Kohsaka, S.; Nakajima, K. Involvement of nitric oxide in the induction of interleukin-1 beta in microglia. Brain Res. 2015, 1625, 121-134.
  3. Xiong, H.; Yamada, K.; Jourdi, H.; Kawamura, M.; Takei, N.; Han, D.; Nabeshima, T.; Nawa, H. Regulation of nerve growth factor release by nitric oxide through cyclic GMP pathway in cortical glial cells. Mol. Pharmacol. 1999, 56, 339-347.

Round 2

Reviewer 1 Report

Comments and Suggestions for Authors

The authors have performed suitable modifications.

Reviewer 2 Report

Comments and Suggestions for Authors

The revised manuscript has been significantly improved. All concerns have been addressed properly. So it can be accepted in the current form.